# Genetic Factors of Nitric Oxide’s System in Psychoneurologic Disorders

**DOI:** 10.3390/ijms21051604

**Published:** 2020-02-26

**Authors:** Regina F. Nasyrova, Polina V. Moskaleva, Elena E. Vaiman, Natalya A. Shnayder, Nataliya L. Blatt, Albert A. Rizvanov

**Affiliations:** 1V.M. Bekhterev National Medical Research Center for Psychiatry and Neurology, Saint-Petersburg, Russian Federation, 192019 Saint-Petersburg, Russianataliashnayder@gmail.com (N.A.S.); 2Institute of Fundamental Medicine and Biology, Kazan Federal University, Kremlevskaya str 18, Kazan 420008, Russia; nataliya.blatt@gmail.com (N.L.B.); rizvanov@gmail.com (A.A.R.); 3Faculty of Medicine and Health Sciences, University of Nottingham, Nottingham NG7 2RD, UK

**Keywords:** NO, NOS, genetics, nitric oxide, nitric oxide synthase, oxidative stress, pathogenesis, mental disorders, neurological diseases

## Abstract

According to the recent data, nitric oxide (NO) is a chemical messenger that mediates functions such as vasodilation and neurotransmission, as well as displaying antimicrobial and antitumoral activities. NO has been implicated in the neurotoxicity associated with stroke and neurodegenerative diseases; neural regulation of smooth muscle, including peristalsis; and penile erections. We searched for full-text English publications from the past 15 years in Pubmed and SNPedia databases using keywords and combined word searches (nitric oxide, single nucleotide variants, single nucleotide polymorphisms, genes). In addition, earlier publications of historical interest were included in the review. In our review, we have summarized information regarding all *NOS1*, *NOS2*, *NOS3*, and *NOS1AP* single nucleotide variants (SNVs) involved in the development of mental disorders and neurological diseases/conditions. The results of the studies we have discussed in this review are contradictory, which might be due to different designs of the studies, small sample sizes in some of them, and different social and geographical characteristics. However, the contribution of genetic and environmental factors has been understudied, which makes this issue increasingly important for researchers as the understanding of these mechanisms can support a search for new approaches to pathogenetic and disease-modifying treatment.

## 1. Introduction

According to the recent data, nitric oxide (NO) is a chemical messenger that mediates functions such as vasodilation and neurotransmission, as well as displaying antimicrobial and antitumoral activities. NO has been implicated in neurotoxicity associated with stroke and neurodegenerative diseases; neural regulation of smooth muscle, including peristalsis; and penile erections [1]. NO is synthesized from L-arginine by macrophages, endothelial cells, neurons, smooth muscle cells, and cardiac myocytes. Three main NO synthase (NOS) isoforms catalyzing the formation of NO have been well characterized. This family includes NOS1, NOS2, and NOS3, which are encoded by *NOS1*, *NOS2*, and *NOS3* genes, respectively [2]. Owing to inducible NOS (iNOS), produced by the gene *NOS2*, high levels of NO are generated to combat environmental insults in a wide range of cells upon induction, while neuronal NOS (nNOS), produced by the gene *NOS1*, and endothelial NOS (eNOS), produced by the gene *NOS3*, control a fluctuating low level of NO to perform normal physiological functions in neurons and vascular endothelial cells [3].

Single nucleotide variants (SNVs) of the NOS gene family are associated with diseases and conditions such as hypertension; QT interval prolongation, a syndrome of sudden cardiac death in children; diabetes mellitus; the risk of miscarriages; and neuropsychiatric disorders. In our review, we have decided to summarize information regarding all *NOS1, NOS2, NOS3,* and *NOS1AP* SNVs involved in the development of mental disorders and neurological diseases/conditions (Table 1, Table 2 and Table 3). We searched for full-text English publications in Pubmed and SNPedia databases from the past 15 years using keywords and combined word searches (nitric oxide, single nucleotide variants, single nucleotide polymorphisms, genes). In addition, earlier publications of historical interest were included in the review. Despite our comprehensive search in these frequently used databases and search terms, some publications might have been overlooked.

## 2. Results

### 2.1. NOS1 Gene

NOS1 is nitric oxide synthase 1, also known as nNOS. However, despite its name, it is expressed not only in the brain, but also in muscle and other tissues in the body (Figure 1).

This enzyme is encoded by the *NOS1* gene located on the long arm of chromosome 12, position 12q24.22 (Figure 2).

#### 2.1.1. Schizophrenia

Shinkai et al. studied the association of the *NOS1* SNVrs2682826 with a risk of schizophrenia. The study involved 215 Japanese patients with schizophrenia and 182 healthy volunteers as a control group. There was a significant difference in the genotype distribution between the patients and the controls (*p* = 0.00122). Moreover, the allele frequency differed significantly between patients with schizophrenia and the controls (*p* = 0.000007; relative risk = 1.92; 95% confidence interval = 1.44–2.55). Thus, the *NOS1* gene was suggested as a candidate gene that increases the susceptibility to schizophrenia [8].

Based on this study, *NOS1* aroused the scientific interest of Fallin et al. They screened 440 SNVs of 64 genes for their associations with a risk of schizophrenia and bipolar affective disorder (BAD) (see Section 2.1.2 on “Depression and Bipolar Affective Disorder” regarding*NOS1*) in 597 (including 323 schizophrenic and 274 BAD) patients and their family members. Only Ashkenazi Jewish families were recruited into the study in order to reduce genetic heterogeneity. Two out of eight *NOS1* SNVs, such as rs3782219 and rs3782221 (*p* = 0.0003 and *p* = 0.0014, respectively), appeared to be associated with schizophrenia. The other six SNVs, including rs2682826, showed no associations [9].

Tang et al. conducted their study in a Chinese population involving 844 patients with schizophrenia (including 425 paranoid schizophrenics) and 861 individuals as a comparative control. Eleven SNVs and one microsatellite (within the exon 1f promoter region) were screened. The study was designed to have twostages. In the first stage, 480 patients and 480 controls were examined; alleles of rs499776 and rs3782206 were found to have associations with a risk of schizophrenia (*p* = 0.014 and *p* = 0.015, respectively), while rs561712 tended to be associated (*p* = 0.054). The second stage involved the examination of these three SNVs, as well as rs3837437, located nearby them in the 5′-flank region of *NOS1*. Here, only rs3782206 demonstrated a statistically significant association, which remained after the correction for multiple comparisons, both in the total sample (*p* = 0.004) and when compared to paranoid schizophrenia (*p* = 0.012). However, all four SNVs were used in haplotype mapping and analysis. Based on these results, a 2-SNV (rs3837437 T–rs3782206 C) haplotype (*p* = 0.0002) was identified as being significantly associated with schizophrenia. There were also statistically significant results for 3- and 4-SNV haplotypes [10].

Reif et al. demonstrated that SNV rs41279104 in the *NOS1* exon 1 promoter region was associated with schizophrenia and prefrontal cortex dysfunctions in a German population consisting of 267 patients with chronic schizophrenia and 284 healthy volunteers [11].

Moskvina et al. reported the association of rs6490121 in the *NOS1* intron 2 with schizophrenia based on thegenome-wide association study (GWAS) results in 479 patients from Great Britain [12].

A group of investigators from Japan tried to reproduce positive results of previous studies conducted in Japanese populations (542 schizophrenic patients and 519 healthy controls at the first stage and 1154 patients and 1260 controls at the second stage of the study). Therefore, seven SNVs were selected. Two SNVs, namely rs3782219 (*p* = 0.0291) and rs3782206 (*p* = 0.0124), had statistically significant allele associations. Interestingly, when the sample size was increased at the second stage, the values were approximately the same: rs3782219 (*p* = 0.0197) and rs3782206 (*p* = 0.0480). However, the results were finally considered erroneous, as they were not significant after the correction for multiple testing [13].

A year later, another group of Japanese investigators repeated the study of their colleagues for the same seven SNVs. The sample size in their study involved 720 subjects (343 schizophrenic patients and 377 healthy controls). According to the genotyping results, only one SNV, rs41279104, which was considered insignificant by the previous authors, demonstrated a statistically significant association with schizophrenia (*p* = 0.006 after the correction for multiple testing). They also studied immunoreactivity in the postmortem brain of patients with schizophrenia. Twelve brain samples collected from patients and 15 samples taken from people without lifetime schizophrenia were examined. Patients with the rs41279104 A-allele had significantly lower NOS1 immunoreactivity levels than GG homozygotes did (*p* = 0.002), which in the authors’ judgement, underlies a significantly decreased NOS1 expression in the prefrontal cortex [14].

Wang et al. evaluated the association between 28 SNVs of the *NOS1* gene in the Chinese Han population (382 patients with schizophrenia and 448 healthy subjects). One SNV rs1520811 showed an association with this disorder; however, it became insignificant after the correction for multiple testing. Thus, their results also do not support the previously identified association between *NOS1* gene polymorphisms and schizophrenia [15].

Weber et al. conducted a meta-analysis of previous studies for genes of the NO system, including *NOS1* and *NOS1AP* (see Section 2.2.1 on“Schizophrenia” regarding *NOS1AP*) and selected eight *NOS1* SNVs for analysis. However, one SNV (rs3837437) was excluded from the analysis as the minor allele frequency was extremely low. This study continued the project by Reif et al. described above. The sample size was extended to 270 German patients with schizophrenia and 720 normal volunteers. Three SNVs, namely rs1879417, rs41279104, and rs499776, showed statistically significant associations. In the meta-analysis, rs41279104 had the best odds ratio (OR = 1.29) [16].

Riley et al. investigated rs6490121 of the *NOS1* gene along with SNVs of other genes and their effects on the development of schizophrenia in an Irish population. A total of 1021 schizophrenic patients and 626 controls were included in the study. There were no associations of the *NOS1* rs6490121 with schizophrenia (*p* = 0.21) [17].

#### 2.1.2. Depression and Bipolar Affective Disorder

Depression and stress are closely related to each other. Stress was shown in animal models to increase the NOS1 expression in many cerebral regions, including the hippocampus [18,19]. Based on this evidence, a group of scientists from Great Britain took an interest in the mutual effect between the most common *NOS1*SNVs, stress factors, and depressive disorders. They examined 1222 subjects. The subjects were genotyped and given questionnaires. Financial hardship was taken as a stress factor. There were significant associations between 8 out of 20 *NOS1* SNVs (rs693534, rs10507279, rs1004356, rs3782218, rs9658281, rs561712, rs522910, and rs2293050) and human liability to depression under exposure to financial and psychosocial stress factors. The results were statistically significant after the correction for multiple testing [20].

Montesanto et al. studied the effect of the most common SNVs in genes of the *NOS* family on a human life span and the quality of life in an old age. *NOS1* SNVs, such as rs1879417 and rs2682826, were screened. A total of 763 individuals (see Section 2.1.6 on “Cognitive Disorders” regarding *NOS1*, *NOS2*, and*NOS3*) were examined. The Geriatric Depression Scale (GDS) was used to evaluate depression levels. The authors demonstrated that subjects with even one minor allele T in SNV rs2682826 had a higher probability of depression symptoms in an old age (*p* = 0.033 after the correction for multiple testing). The second studied SNV (rs1879417) had no statistically significant effect on depressive disorders; however, it was associated with maintaining cognitive abilities with age [21].

Winger et al. studied the association between polymorphisms of the genes related to oxidative and nitrosative stresses, including *NOS1* and *NOS2* (see Section 2.3 on “*NOS2 Gene*”), and a risk of depression in a Polish population (281 depressed patients and 229 controls). The frequency of the study *NOS1* SNV (rs1879417) did not show significant differences between samples [22].

Also, a group of Japanese scientists (Okumura et al.) investigated an association between the *NOS1* SNV rs41279104 and mood disorders. In addition to depression, they studied BAD. This SNV was chosen as plasma NO levels had been previously reported to change in patients with mood disorders (a decrease in depressed patients [23] and an increase in type I BAD patients compared to the control [24]). The study included 325 patients with depression, 154 patients with BAD, and 807 healthy volunteers. However, there were no statistically significant associations with depression or BAD [25].

Fallin et al. found no significant associations between BAD and the *NOS1*SNV in their study (see Section 2.2.1 on “Schizophrenia” regarding *NOS1*) [9]. Buttenschon et al., the authors of the study preceding that conducted by Fallin and who prompted the inclusion of this gene in the project, conducted their study in two populations: British and Danish. No significant differences in the frequencies of SNV rs2682826 were observed among all the subjects. However, there was a difference in genotypes between the Danish and control groups (*p* = 0.045). Yet, the investigators considered it incidental, as the Danish population was smaller (83 patients) than the British one (286 patients) and there was no allelic association [26].

#### 2.1.3. Autism Spectrum Disorders

Unlike *NOS2*, the association study of *NOS1* and *NOS2* genes (see Section 2.3.2 “Autism Spectrum Disorders” regarding *NOS2*) with autism spectrum disorders (ASD) in an Asian population demonstrated no statistically significant results with respect to*NOS1*. Nine *NOS1* SNVs, namely rs2682286, rs2293044, rs3741475, rs3741476, rs1047735, rs2293054, rs3741480, rs9658255, and rs9658247, were analyzed [27].

#### 2.1.4. Parkinson’s Disease

As mentioned above, high concentrations of NO produce neurotoxic effects in the body. The balance between useful and toxic properties of NO in nerve cells (in particular, neurons of the substantia nigra) varies both under exposure to environmental factors and depending on the expression levels of the genes responsible for nitrogen synthesis. Therefore, these genes are candidates for increasing the risk of Parkinson’s disease (PD).

Hancock et al. screened 1065 patients with PD for *NOS1*, *NOS2A*, and *NOS3* SNVs, showing they increase the risk of the disease (see Section 2.3.3 and Section 2.4.4 on “Parkinson’s Disease” regarding *NOS2* and *NOS3*), as well as the gene interaction with the environment (cigarette smoking, use of caffeine, nonsteroidal anti-inflammatory drugs, and pesticides). Twenty-seven SNVs of the *NOS1* gene were included in the analysis. There were significant associations between PD and SNVs rs12829185, rs1047735, rs2682826, rs3782218, rs11068447, rs7295972, rs2293052, and rs3741475 (with a range of *p* = 0.00083–0.046). Furthermore, the first three of these SNVs demonstrated an interaction with pesticides (with a range of *p* = 0.012–0.034) [3].

#### 2.1.5. Brain Tumors

A group of American scientists led by Bhatti studied lead exposure (as an environmental factor) in terms of its involvement in mechanisms of neurotoxicity and the genetic aspects of these interactions. Oxidative stress is a component of neurotoxicity (particularly with lead intoxication). For example, lead was shown to deplete antioxidant proteins and to stimulate the production of reactive oxygen species [28,29]. A total of 496 patients (with 362 gliomas including 176 glioblastomas and 134 meningiomas) were included in the study (see Section 2.3.5 and Section 2.4.8 “Brain Tumors” regarding*NOS1* and *NOS2*). At first, 11 *NOS1* SNVs demonstrated a statistically significant association (*p* ≤ 0.05) with one or more types of cerebral tumors or a significant modification of lead cumulative effects on brain tumors. However, none of the associations was significant after the correction for multiple testing [30].

#### 2.1.6. Cognitive Disorders

The above study (see Section 2.1.2 on “Depression and Bipolar Affective Disorder” regarding *NOS1*) also included the measurement of the subjects’ cognitive functions with the use of the MMSE (mini mental state examination) scale. The C minor allele of SNV rs1879417 was at first an associated with low cognitive characteristics, i.e., the lowest MMSE scores (*p* = 0.045). However, after the correction for multiple testing, this association was deemed statistically insignificant (*p* = 0.093) [21].

However, we have foundseveral studies that confirm the role of *NOS1*SNV in the risk of cognitive impairment in neuropsychiatric diseases, particularly in schizophrenia, based on previously shown associations of some SNVs and schizophrenia. To date, it is not clear which of these factors are primary and which ones are secondary. Cognitive impairments in tasks involving the prefrontal cortex (for example, working memory or verbal fluency) are key in schizophrenia. This led to the hypofrontality hypothesis of schizophrenia. The impairment of glutamatergic neurotransmission is thought to certainly play a role. The promoter region of *NOS1*, affecting the prefrontal transmission of glutamate, has been repeatedly associated with schizophrenia.

A group of Irish scientists studied rs6490121. In 2009, they conducted a large study involving 349 Irish patients with schizophrenia, 232 German patients, and a control group. The Irish were first examined, then the results were confirmed in the second population. G allele carriers among patients showed poorer verbal intelligence quotient (IQ) and working memory test results [31]. Studies of this SNV were further continued. O’Donoghue et al. demonstrated its participation in early sensory processing in 54 healthy subjects. G allele carriers had lower electroencephalogram (EEG) responses to visually evoked potentials P1 [32]. Rose et al. reported structural and functional changes in the prefrontal cortex in G allele carriers [33].

The number of studies suggesting impaired prefrontal functioning in schizophrenia continue to increase and there is increasingly more evidence of the genetic variation of *NOS1* in cognitive dysfunction, probably by decreasing glutamatergic transmission. Reif et al. sought for associations between another SNV rs41279104, which leads to a decreased transcript expression, schizophrenia (see Section 2.1.1 on “Schizophrenia” regarding*NOS1*), and the prefrontal cortex functioning. The study involved 87 subjects (43 patients with schizophrenia and 44 controls). In addition to genotyping, functional spectroscopy was performed with simultaneous working memory testing. Task-related changes in oxygenation were significantly reduced in patients with schizophrenia. Schizophrenic patients (A allele carriers) showed the worst results [34].

Zhang et al. summarized previous studies and repeated them with respect to another SNV rs3782206. They showed that the schizophrenia risk allele (T) of rs3782206 was associated with the poorest results in five measures of cognitive performance deficiency in patients (580 patients with schizophrenia) and only in three patients among the controls (720 healthy volunteers). Functional spectroscopy revealed reduced activation in the right inferior frontal gyrus of the risk allele carriers during cognitive testing [35].

These results strongly suggested an association between *NOS1* gene variants and cognitive functions, as well as their neural underpinnings; they have important implications for our understanding of the neural mechanism underlying the association between *NOS1* SNVs and schizophrenia.

Research in neurooncology has also been conducted in parallel with a more comprehensive study of associations with schizophrenia. Liu et al. conducted a large exploratory study of 10,967 SNVs in 580 genes, including *NOS1*, in 233 newly diagnosed glioma patients before surgery. The strongest associations with respect to NOS1 were in executive function testing and rs11611788 (*p* = 0.000000018) [36].

#### 2.1.7. Ischemic Stroke

NO, produced by endothelial cells, causes vasodilation and hypotension; it has several anti-thrombotic and anti-atherosclerotic properties as well [37]. Therefore, NOS, neuronal NOS in particular, plays a pivotal role in the development of atherosclerosis and the regulation of blood flow, and it is likely that its effect on ischemic stroke (IS) might be mediated by these two main clinical risk factors.

For example, Manso et al. investigated the effects of *NOS1* and *NOS3*SNVs on IS susceptibility and outcome after an IS (see Section 2.4.5 on “Ischemic Stroke” regarding*NOS3*). The study population consisted of 551 IS patients and 530 controls. Thirty-seven*NOS1* SNVs were included in the analysis. Four SNVs (rs2293050, rs2139733, rs7308402, and rs1483757) were significantly associated with IS susceptibility (a range of *p* = 0.036–0.048 after being corrected for multiple testing); NOS1 variants were not associated with IS outcomes [38].

Dai et al. conducted a study with a similar design (a case-control study) in a Chinese population. A total of 413 patients with IS and 477 healthy individuals were examined. It was shown that only rs7308402 was associated with an increased risk of IS (the A allele is a protective factor: genotype AG, *p* = 0. 037; allele A, *p* = 0.041 compared to the control). Unlike in the study conducted by Manso, rs1483757, rs2139733, and rs2293050 did not display statistically significant differences [39].

#### 2.1.8. Restless Legs Syndrome

Winkelmann et al. performed a three-stage study. The first stage involved screening for 1536 SNVs in 366 genes. At the second stage, the most significant SNVs of stage 1 were genotyped. Significant associations were observed with *NOS1* rs7977109 and rs693534 in both the explorative and replication study stages. However, the same alleles were protective in the former, while being a risk factor in the latter. The investigators suggested this might be due to general differences between certain samples, rather than the *NOS1* association with restless legs syndrome (RLS). They tested this hypothesis using the method of genomic control. The discrepancy in the results between two samples may alternatively be accounted for by the interaction of *NOS* with environmental factors. This seems possible, given that NOS is involved in the metabolism of arginine, and its levels may differ due to eating habits. Finally, differences can also be accidental. The revision was performed only in one replication sample. In conclusion, the investigators do not consider that opposite associations should be excluded from further analysis and research, since they may well indicate a true relationship. However, further research in independent populations is required [40].

#### 2.1.9. Multiple Sclerosis

The association study of NO-encoding genes with multiple sclerosis (MS) demonstrated no statistically significant results in relation to the *NOS1* gene, unlike other enzymes of this family (see Section 2.3.7 and Section 2.4.10 on “Multiple Sclerosis” regarding*NOS2* and *NOS3*). SNVs rs2682826 and rs41279104 were included in the analysis [41].

### 2.2. NOS1AP Gene

When discussing nNOS, it is necessary to note its adaptor protein. NOS1 (neuronal) adaptor protein (NOS1AP) is a cytosolic protein that binds to the signaling molecule, nNOS. The protein has a C-terminal PDZ-binding domain that mediates interactions with nNOS, and a N-terminal PTB-domain, that binds to the small monomeric G-protein, DEXRAS1.

The expression of NOS1A P is given in Figure 3.

This enzyme is encoded by the *NOS1AP* gene located on the long arm of chromosome 1, position 1q23.3 (Figure 4). Its original name, *CAPON*, is also encountered in the literature.

#### 2.2.1. Schizophrenia

As NOS1AP is involved in signal transmission in the system of N-methyl-d-aspartate-receptors (NMDAR), it is a potentiallyimportant component in the etiology of schizophrenia.

A group of American scientists studied associations between SNVs of this gene and the risk of schizophrenia. Twenty-four Canadian families participated in the study. Brzustowicz et al. screened for 15 SNVs (rs1572495, rs1538018, rs945713, rs1415263, rs4306106, rs3934467, rs3924139, rs4145621, ss16342089, rs2661818, rs1508263, rs3751284, rs7521206, rs348624, and ss16342088). Out of them, six SNVs had statistically significant associations (rs1572495: *p* = 0.021; rs1538018: *p* = 0.047; rs945713: *p* = 0.016; rs1415263: *p* = 0.0016; rs4145621: *p* = 0.0016; rs2661818: *p* = 0.0032) [42]. Wratten et al. continued this project and extended the number of SNVs up to 38. They showed alleles C of rs1415263, T of rs4145621, and A of rs12742393 to be associated with schizophrenia, which acts by enhancing transcription factor binding and increasing gene expression. Thus, one more SNV, namelyrs12742393, demonstrated significant differences in allele expression. Allelic variation in this SNV changed the affinity of a core protein that binds to this deoxyribonucleic acid (DNA) region. Therefore, the authors suggested the A allele of rs12742393 to be possibly a risk allele associated with schizophrenia [43].

A study was performed in an Asian population. Zheng et al. studied nine *NOS1AP* SNVs (rs1572495, rs945713, rs1415263, rs4145621, rs2661818, rs3751284, rs905721, rs348624, and rs1964052); rs2275643 and rs11422090 were considered but excluded during the study. A study sample consisted of 664 patients with schizophrenia and 941 controls of the Chinese Han population. One SNV demonstrated statistically significant differences in the analysis of both allele (*p* = 0.000017) and genotype (*p* = 0.000030) frequencies, and haplotypes with this SNV (rs905721, rs348624, rs1964052 – *p* = 0.000025). This study is one of those that provide support for the potential importance of NMDAR-mediated glutamatergic transmission in the etiology of schizophrenia [44].

Thirteen *NOS1AP*SNVs (rs1572495, rs1538018, rs945713, rs1415263, rs4306106, rs3924139, rs1508263, rs3751284, rs7521206, rs905721, rs348624, rs1964052, and rs4145621) were analyzedin the above study performed by Weber et al. (see Section 2.1.1 on “Schizophrenia” regarding*NOS1*). SNV rs4145621 was excluded during the analysis. Six SNVs of *NOS1AP* were significantly associated with schizophrenia in at least one population (German, Swedish, Spanish): rs945713 (*p* = 0.002—German), rs1415263 (*p* = 0.006—Swedish), rs4306106 (*p* = 0.018—Swedish), rs3924139 (*p* = 0.003 – Swedish), rs1508263 (*p* = 0.006—Spanish), and rs3751284 (genotype *p* = 0.022—Swedish). Notably, rs945713 was statistically significant after the Bonferroni correction in the German sample (*p* = 0.048) and demonstrated a tendency toward association in the Swedish sample (*p* = 0.064) [16].

#### 2.2.2. Post-Traumatic Stress Disorder

As discussed above, depression and stress are two causal factors that are inseparably related to each other. The NOS1 adaptor protein modulates stress-evoked N-methyl-d-aspartate (NMDA)activity. Post-traumatic stress disorder (PTSD) is an anxiety-depressive disorder that debuts after exposure to a traumatic event. PTSD is most common among combat veterans. Lawford et al. screened for associations between this disorder and *NOS1AP* SNVs in 121 Vietnam combat veterans and 237 healthy Caucasian volunteers. PTSD patients were assessed for symptom severity and depression levels using the Mississippi Scale for Combat-Related PTSD and the Beck Depression Inventory-II (BDI). Thirteen SNVs (rs945706, rs1415259, rs4656355, rs6704393, rs1415263, rs4531275, rs6683968, rs4657178, rs2341744, rs347300, rs1858232, rs347313, and rs386231) were investigated in the study. The G allele of rs386231 appeared to be significantly associated with PTSD (*p* = 0.002). There were reliable data that the GG genotype increased the severity of depression (*p* = 0.002 F = 6.839) and had a higher Mississippi Scale for Combat-Related PTSD scores (*p* = 0.033). The haplotype analysis revealed that the C/G haplotype (rs451275/rs386231) was associated with PTSD (*p* = 0.001). However, the authors note that their study is one of the first in this direction and that their sample sizes were not sufficient to detect SNV associations with very small effects. Nevertheless, they suggest that the *NOS1AP* SNV rs386231 may increase susceptibility to severe depression in patients with PTSD and thereby increase a risk forsuicide [45].

### 2.3. NOS2 Gene

The *NOS2* gene encodes nitric oxide synthase 2, which is expressed in the liver and induced by a combination of lipopolysaccharide and certain cytokines. This radical is encoded by the gene *NOS2* (OMIM:163730) on the long arm of chromosome 17 (position 17q11.2) (Figure 5). The gene is also known as *NOS*, *INOS*, *NOS2A*, and *HEP-NOS* [7].

*NOS2* is mainly expressed in the small and large intestine, kidneys, liver, and lungs (Figure 6) [5].

#### 2.3.1. Depression

There is certain evidence of the imbalance in the generation and elimination of reactive oxygen (ROS) and nitrogen (RNS) species in depression [46]. This imbalance results in increased levels of intensified oxidative and nitrosative stress biomarkers, such as 8-hydroxyguanine (8-oxoG), 8-iso prostaglandin F2a (8-iso-PGF2a), malondialdehyde (MDA), and NO [47,48]. Moreover, patients with depression have an increased expression of cellular NOS in neurons of the suprachiasmatic nucleus compared to the control group. The excessive activity ofa prooxidant enzyme NOS result in increased ROS and RNS levels that can lead to neurodegenerative changes [49,50]. Wigner et al. investigated SNVs of *SOD2, CAT, GPx4*, and *NOS1* genes (see Section 2.1.2 on “Depression and Bipolar Affective Disorder” regarding *NOS1*)and *NOS2* SNVs c.−227G>C(rs10459953) and c.1823C>T(p.Ser608Leu)(rs2297518) in 281 depressed patients in comparison with 299 controls of the Polish population. The analysis revealed no association of a risk of depression with SNVs c.−227G > C(rs10459953) and c.1823C > T (p.Ser608Leu)(rs2297518) in *NOS2*. No correlation between haplotypes of these SNVs was found. However, the gene–gene analyses revealed that a risk of depression increased five-fold with genotypes such as T/T-T/T of *SOD2* c.47T > C(p.Val16Ala)(rs4880) and *NOS2* c.1823C > T(p.Ser608Leu)(rs2297518) polymorphisms (*p* = 0.013). The risk of depression increased twofold with combined T/T-T/Tgenotypes of *NOS2* c1823C > T(rs2297518) and *GPx4* c. 660T > C(rs713041) polymorphisms (*p* < 0.001). There was also a correlation between genotypes such as G/C-T/T (*p* = 0.001) and G/G-T/T (*p*=0.015) of *NOS2* SNV c.- 227G > C(rs10459953) and *GPx4 SNV*c.660T > C(rs713041) with depression. T/T-C/T genotypes of SNVs c.−89A > T(rs7943316) in *CAT* and c.1823C > T(rs2297518) in *NOS2* increased the risk of depression (*p* = 0.002), while the T/T-T/T genotype reduced the risk (*p* = 0.036). The A/T-G/G genotype of c.−89A > T(rs7943316) in *CAT* and c.−227G > C(rs10459953) in *NOS2* was associated with a risk of depression (*p* = 0.001) [22].

#### 2.3.2. Autism Spectrum Disorders

NO is an important signaling molecule that is involved in the development of the central nervous system (CNS) and certain physiological functions, such as the release of noradrenaline and dopamine, which affect memory and learning. It is also involved in the development of BAD, schizophrenia, and depression [51,52]. In the brain, the *NOS2* gene is found in activated immune cells, such as the microglia and astroglia. It is involved in the demyelination of the central nervous system and neuronal death. Patients with ASD have immune and inflammation diseases, including T-, B-, and NK-cell dysfunction and increased levels of pro-inflammatory cytokines [53]. Since NO plays an important role in neuroinflammation, it has been proposed that it be considered in the pathogenesis of ASD [27]. Kim et al. conducted a genetic study in 151 women with ASD, where 9 *NOS1* SNVs (see Section 2.1.3 on “Autism Spectrum Disorders” regarding *NOS1*) and 9 *NOS2*SNVs (rs7406657, rs3201742, rs2255929, rs8068149, rs1060826, rs2297518, rs1137933, rs10459953, and rs2779248) were analyzed. The analysis results demonstrated that the allele A of SNV rs8068149 (*p* = 0.039) and allele G of SNV rs1060826 (*p* = 0.035) in *NOS2* were associated with a risk of ASD. Moreover, a risk of ASD was higher for GG or AG genotypes than for that of AA of *NOS2* SNV rs1060826 [27].

#### 2.3.3. Parkinson’s Disease

Hancock et al. studied associations of 27 *NOS1* SNVs (see Section 2.1.4 on “Parkinson’s Disease” regarding*NOS1*), 18 *NOS2* SNVs (rs3730014, rs3794766, rs2072324, rs8072199, rs16966563, rs3794764, rs17722851, rs944725, rs4795067, rs1137933, rs12944039, rs2314810, rs2248814, rs2297518, rs2297516, rs2297515, rs1060826, and rs2255929) and 5 *NOS3*SNVs (see Section 2.4.4 on “Parkinson’s Disease” regarding*NOS3*) with a risk for PD in 337 families with sporadic PD from the USA and 358 families with familial PD. The analysis results showed that the A allele of rs2072324, A allele of rs3794764, G allele of rs12944039 (*p* < 0.0001), A allele of rs2297516 (*p* < 0.0001), and T allele of rs2255929 (*p* < 0.0001) of the *NOS2* gene were associated with a risk for PD in 337 families with sporadic PD at ages between 40 and 80. However, no association of *NOS2* with familial PD was found. The carrier status of the *NOS2* SNV rs2255929 was associated with PD in patients younger than 40 years old (*p* = 0.046) [3].

#### 2.3.4. Migraines

The etiology of migraines is based on a neurogenic inflammatory component that affects blood vessels by enhancing the formation of NO with subsequent pain. Excessive NO amounts are possibly derived from increased *NOS2* expression and activity [54].

De OS Mansur et al. tested two potentially functional clinically relevant SNVs, namely G2087A (rs2297518) and C(−1026)A (rs2779249), of *NOS2* in 148 women with migraines with auras and 52 women with migraine without auras in comparison with the control group consisting of 152 healthy women from Brazil. It was found that the carrier status of the *NOS2* SNV G2087A (rs2297518) was associated with a risk of migraine. Notably, the A allele of G2087A SNV was more common in the group of patients with migraines with auras than in those without auras (*p* = 0.0243). The AA genotype for *NOS2* SNV G2087A (rs2297518) and C(−1026)A (rs2779249) was associated with migraines with auras(*p* = 0.0349)[54].

Gonçalves et al. studied SNVs of *NOS2* C−1026A (rs2779249) and G2087A (rs2297518), *NOS1* (see Section 2.4.7 on “Migraines” regarding *NOS3*) and *VEGF* in 150 women with migraines in comparison with the control group consisting of 99 healthy women from Brazil. The analysis results showed that *NOS2* SNV G2087A (rs2297518) was associated with a risk of migraines (*p* = 0.0120). At the same time a similar association of *NOS2* SNV C(−1026)A (rs2779249) was not found [55].

#### 2.3.5. Brain Tumors

Bhatti et al. studied the associations of nine *NOS1* SNVs (see Section 2.1.5 on “Brain Tumors” regarding*NOS1*); *NOS2* SNVs rs944725, rs4795067, rs2297516, rs2779252, and rs8072199; and three*NOS3* SNVs *(*see Section 2.4.8 on “Brain Tumors” regarding*NOS3)* with a risk of brain tumors in 362 patients with glioma (176 of whom had glioblastoma multiforme), 134 patients with meningioma, and 494 healthy controls. There is statistically significant evidence that the carrier status of *NOS2* SNVs rs944725, rs2779252, and rs8072199 was associated with a risk of meningioma (*p* = 0.03), SNV rs4795067 was associated with a risk of glioblastoma (*p* = 0.05) and meningioma (*p* = 0.02), while SNV rs2297516 (*p* = 0.05) was associated with a risk of glioma [30].

#### 2.3.6. Ischemic Stroke

Yan et al. studied 588 patients with IS and 557 healthy controls for associations of SNVs of *NOS2* Leu608Ser (rs2297518), *NOS3* (see Section 2.4.5 on “Ischemic Stroke” regarding *NOS3*), and*GCH1* and *CYBA* with a risk of IS. The analysis results showed no statistically significant associations of *NOS2* SNV G>A Leu608Ser (rs2297518) with a risk of IS when studying a Chinese Han population [56].

Wang et al. investigated SNVs of *NOS2* 231C>T(rs1137933) and *NOS3* (see Section 2.4.5 on “Ischemic Stroke” regarding *NOS3*). The meta-analysis of six studies performed in the USA, Europe, and China did not obtain statistically significant results in 3550 patients and 6560 controls [57].

#### 2.3.7. Multiple Sclerosis

Al Fadhli et al. screened 122 patients with MS and 188 healthy individuals for SNVs of *NOS1* (see Section 2.1.9 on “Multiple Sclerosis” regarding *NOS1*), *NOS2* (CCTTT)n/(TAAA)n, and *NOS3*(see Section 2.4.10 on “Multiple Sclerosis” regarding *NOS3*). The analysis results showed the association of the *NOS2* (CCTTT)n/(TAAA)n genotype with a risk of MS in a population from Kuwait [41].

### 2.4. NOS3 Gene

The *NOS3* gene encodes nitric oxide synthase 3*. NOS3,* also known as *eNOS* and *ECNOS,* is localized on the long arm of chromosome 7, position 7q36.1 (Figure 7). Its expression is shown in Figure 8.

#### 2.4.1. Methamphetamine-Induced Psychosis

Methamphetamine (METH) is an illegal addictive drug that causes mental disorders. According to recent findings, METH selectively increases the NO concentration in the (corpus) striatum, which leads to dopaminergic neurotoxicity in the human brain. Okochi et al. studied functional SNVs (rs1800779, rs2070744, rs1799983, rs3918188, rs743507, and rs7830) of *NOS3* in 183 patients with METH-induced psychosis (METH-i.ps.) in comparison with 267 controls. No significant association was found in the study between*NOS3* SNVs and the risk of METH-i.ps in the Japanese population [58].

#### 2.4.2. Depression

There is evidence that depression is related to an increased risk of mortality and morbidity fromcoronary artery disease (CAD). Three *NOS3* SNVs, namely rs2070744, rs1799983, and VNTR, are reported to be associated with CAD. Ikenouchi-Sugita et al. studied these SNVs in 51 depressed patients in the Japanese population. The study results suggest no association between the study SNVs and a risk of depression [59].

#### 2.4.3. Suicide

The *NOS3* gene is involved in the proliferation of neuronal precursors that might be associated with the pathology of depressive disorders. *NOS3* SNVs T-786C (rs2070744) and 27bpVNTR are considered to be associated with a risk of BAD and suicidal behavior. Sáizet al. found no statistically significant results in 186 suicide attempters compared to a group of 420 healthy controls from the Spanish population [60].

#### 2.4.4. Parkinson’s Disease

Hancock et al. studied the associations of 27 *NOS1* SNVs (see Section 2.1.4 on “Parkinson’s Disease” regarding*NOS1*); 18 *NOS2* SNVs (see Section 2.3.3 on “Parkinson’s Disease” regarding*NOS2*); and 5 *NOS3* SNVs, namely rs1800783, rs1549758, rs1799983, rs3918227, and rs1808593, with a risk of PD in 337 families with sporadic PD from the USA and 358 families with familial PD. The study found no association of *NOS2* with PD in patients within the ages of 40 to 80. However, the carrier status of *NOS3*SNV rs1808593 was associated with PD in patients younger than 40 years old (*p* = 0.046) [3].

#### 2.4.5. Ischemic Stroke

Yan et al. investigated the association of *NOS2* SNVs (see Section 2.3.6 on “Ischemic Stroke” regarding*NOS2*)*,* and *NOS3* SNVs G>T(D298E)(rs1799983) andT>C(rs2070744) with a risk of IS in 558 patients with IS and 557 healthy controls. No associations with the studied SNVs were found [56].

In a meta-analysis of six studies performed in the USA, Europe, and China, Wang et al. analyzed the association of *NOS2*SNVs (see Section 2.3.6 on “Ischemic Stroke” regarding *NOS2*) and *NOS3*SNVs (−922)A>G(rs1800779), (−690)C>T(rs3918226), and 298Glu>Asp (rs1799983) with a risk of IS in 3550 patients after an ischemic stroke and 6560 controls;no associations were detected [57].

Du et al. found that the A allele of *NOS3*SNV rs3918181 (*p* = 0.684) was associated with a risk of IS in men in a sample of 560 patients with IS and 153 healthy controls in a Chinese population. At the same time, this association was not observed among women [61].

Manso et al. investigated the effect of *NOS1*SNVs (see Section 2.1.7 on “Ischemic Stroke” regarding*NOS1*) and *NOS3* SNVs rs1800783 and rs2373929 on IS susceptibility and the outcome after IS. The study population consisted of 551 stroke patients and 530 controls. The study results revealed no statistically significant evidence that supported the association of the studied SNVs and a risk of IS [38].

#### 2.4.6. Dementia

Dementia develops in 25% of patients who have suffered an IS. The *NOS3* gene is one of those associated with vascular regulation; however, it has not been considered as a potential risk factor for dementia. In the study performed by Morris et al., the TT genotype of *NOS3* SNV p.Asp298Glu (rs1799983)*(p* = 0.001) increased the risk of dementia compared to the GG genotype in the study population of 253 post-stroke patients older than 75 years [62].

#### 2.4.7. Migraines

Schürks et al. studied *NOS3* SNVs rs1800779, rs3918226, and rs1799983 in 4705 women with migraines and 21,008 healthy women. The analysis results supported the association of *NOS3*SNV rs3918226 with the risk of migraines without auras (*p* = 0.04) [63].

The study of Toriello et al. did not confirm the association of *NOS3* SNVs 786T>C(rs1800779) and Glu298Asp (rs1799983) with migraines in 337 patients, including 188 migraines with auras, and 341 healthy individuals from the Spanish population [64].

Gonçalves et al. investigated *NOS3* SNVs T(−786)C (rs2070744), Glu298Asp (rs1799983), 27 bpvariable number of tandem repeats (VNTR), rs3918226, and rs743506 in 178 women with migraines, including 44 patients with auras and 134 patients without auras. None of the five SNVs was statistically significant. The GA genotype of *NOS3* SNVrs743506 was more common in the control group than in patients with migraines with auras. However, haplotypes CCaGluG and “CCbGluG” of the *NOS3* gene (*p* < 0.0015) were associated with migraines with auras [65].

#### 2.4.8. Brain Tumors

Bhatti et al. screened SNVs of *NOS1* (see Section 2.1.5 on “Brain Tumors” regarding*NOS1*)*; NOS2* (seeSection 2.3.5 on “Brain Tumors” regarding*NOS2*)*,* and *NOS3* rs1799983, rs4496877, and rs12703107 for a risk of brain tumors. Statistically significant findings indicate that the carrier status of SNV rs1799983 (*p* = 0.04) was associated with the risk of glioblastoma multiforme, rs12703107 was associated with glioblastoma multiforme (*p* = 0.007) and glioma (*p* = 0.04), while rs4496877 was associated with meningioma (*p* = 0.02) [30].

#### 2.4.9. Infantile Cerebral Palsy

Asphyxia, neurological diseases, infections in the uterine cavity, amnionitis, maternal autoimmune diseases, metabolic disorders, and vascular lesions are risk factors for infantile cerebral palsy (ICP). Genetic theory has been recently considered and *NOS3* is one of the candidate genes. Wu et al. performed a meta-analysisof 11 studies and found no association of *NOS3* SNVs rs1800779, rs1799983, and rs3918226 with a risk of ICP in 2533 patients with ICP in comparison with the control group consisting of 4432 healthy children [66].

#### 2.4.10. Multiple Sclerosis

Al Fadhli et al. studied the SNVs of *NOS1* (see Section 2.1.9 on “Multiple Sclerosis” regarding*NOS1*), *NOS2* (see Section 2.3.7 on “Multiple Sclerosis” regarding*NOS2*), and *NOS3* (rs1800783, rs1800779, rs2070744, and 27bpVNTR). The analysis results demonstrated that the G allele of *NOS3*SNV rs1800779 (*p* = 0.04) and the GG genotype (*p* = 0.02) were associated with a risk of MS. The A allele of *NOS3* SNV 27bpVNTR was associated with an early onset of MS (≤26 years old, *p* = 0.043). The A/b haplotype of *NOS3* SNV 27bpVNTR leads to a 23% decrease of NO production, while the *NOS3* expression decreased with the AA genotype of the gene SNV rs1800779 [41].

#### 2.4.11. Gentamicin-Induced Vestibular Dysfunction

The use of aminoglycoside antibiotics, such as gentamicin (GM), is known to lead to permanent ototoxicity. NOS inhibition has been shown to reduce the toxicity of GM. Roth et al. studied candidate genes that promote susceptibility to GM-induced vestibular dysfunction(GM-i.v.d.). A proposed oxidative stress’s model of GM-induced ototoxicity underlay the selection of these candidate genes. The authors studied *NOS3* SNVs c.893G>T(p.Glu298Asp)(rs1799983), c.−813T>C(rs10952298), and c.582+250N27 (4_5)** VNTR in 137 patients with unilateral or bilateral GM-i.v.d. and 126 healthy controls from the American population. It was found that a carrier status for SNV c.893G>T(p.Glu298Asp)(rs1799983) was associated with a risk of GM-i.v.d. (*p* = 0.03) [67].

#### 2.4.12. Hypoxic-Ischemic Encephalopathy (HIE)

The blood–brain barrier permeability is impaired in hypoxic-ischemic encephalopathy (HIE). This process is multifactorial and results from oxidative stress, increased vascular endothelial growth factor levels, and increased inflammatory cytokines and NO concentrations. *NOS3* is predominantly expressed in vascular endothelial cells and can prevent neuronal injury by producing small amounts of NO to expand blood vessels, maintain cerebral blood flow, inhibit platelet aggregation, and prevent oxidative damage. Wu et al. studied *NOS3* SNVs rs1800783, rs1800779, and rs2070744 in 226 children with HIE and 212 healthy children with a birth weight of 1001–1449 g in a Chinese population. Apgar scores and magnetic resonance image scans were used to estimate the symptoms and brain damage. According to the results, the distribution of *NOS3* SNV rs2070744 significantly varied in children with different Apgar scores (≤5, TT/TC/CC = 6/7/5; 6–7, TT/TC/CC = 17/0/0; 8–9, TT/TC/CC = 6/2/0; 10, TT/TC/CC = 7/1/0; *p* = 0.006). Thus*,* the *NOS3* SNV rs2070744 (*p* = 0.026) was associated with a high susceptibility to HIE [68].

Kuzmanić Samija et al. analyzed *NOS3* SNVs rs3918186, rs3918188, rs1800783, rs1808593, rs3918227, rs1799983, and rs1800779 in the Croatian population consisting of 110 preterm children with HIE and 128 preterm children without HIE at the age of 2 years and older. Genotyping results showed that a risk of HIE was only shown in association with SNVrs1808593 (*p* = 0.0023). At the same time, the TG haplotype of rs1800783-rs1800779 (*p* < 0.00) also showed an association with HIE [69].

#### 2.4.13. Cerebral Ischemia Following Subarachnoid Hemorrhage

Endothelial dysfunction, pro-inflammatory processes in the vascular bed, and an impaired fibrinolytic cascade following aneurysmal subarachnoid hemorrhage (aSAH) due to aneurysm rupture can contribute to a cerebral microthromboembolism. A massive cerebral vasospasm reduces the cerebral blood flow and increases the risk of a cerebrovascular accident. NO is a key molecule for maintaining the cerebral blood flow and regulating platelet aggregation, white blood cell adhesion and migration, and smooth muscle proliferation. Results of the study performed by Hendrix et al. over 4 years (2012–2015) demonstrated the C allele of *NOS3* SNV rs2070744 (*p* = 0.040) increased the risk for a delayed cerebral ischemia in a group of 149 patients with aSAH. At the same time, this SNV was not associated with a functional outcome or the size of the aneurysm at the time of rupture [70].

## 3. Conclusions

Neuropsychiatric disorders are multifactorial in nature; that is, both endogenous and exogenous factors are the basis of their development. From this point of view, the NO system is of significant interest. Considering the existing theories of the development of neuropsychiatric disorders (e.g., hypoxia, impaired glutamatergic neurotransmission, excitotoxicity, oxidative and nitrosative stress, thrombosis and atherosclerosis, and vasospasms), NO, being directly involved in all of the chemical processes, is also involved in their biological manifestation.

NOS are expressed by many cell types of the human body, which explains the diversity of their functions.

nNOS are expressed mainly in central and peripheral neurons, as well as some other cell types, providing a synaptic plasticity in the central nervous system, central regulation of blood pressure, relaxation of smooth muscles, and vasodilation through peripheral nitrergic nerves.iNOS are expressed in many cells in response to lipopolysaccharides, cytokines, or other agents. iNOS generates a large amount of NO, which has a cytostatic effect on target cells of microorganisms, thereby underlining the pathophysiology of inflammatory diseases and septic shock, and are also involved in autoimmune reactions.eNOS are predominantly expressed in endothelial cells. They ensure the expansion of blood vessels, control blood pressure, and are responsible for other vasoprotective and anti-atherosclerotic effects. Many cardiovascular risk factors lead to oxidative stress and endothelial dysfunction in the vasculature.

An analysis of currently available literature demonstrated existing scientific and clinical interest in the prognostic role of the polymorphisms of gene encoding enzymes involved in the synthesis of NO. However, to date, the contribution of genetic and environmental factors has not been sufficiently studied. Understanding these mechanisms can help to find new approaches to pathogenetic and disease-modifying treatments.

This literature review indicates that genetic factors have a significant impact on the development of neuropsychiatric disorders. The existing contradictions can be explained in terms of different study designs, small sample sizes, and various socio-geographical and ethnic factors. In order to achieve results with a high level of evidence, it is necessary to carry out inter-center projects to provide larger samples and take into account the ethnicity of the participants. Environmental considerations should also be an integral part of every study. Epigenetic methods that reflect the influence of various factors on gene expression should be included in such projects to obtain multilateral conclusions regarding oxidative stress in the pathogenesis of neuropsychiatric disorders.

To date, a lack of comprehensive understanding of the genetic predisposition and multifactorial nature of diseases does not allow us to recommend an etiological treatment. Nevertheless, from the point of view of pathogenetic disease-modifying therapy, the study of the NO system is a promising area. In this regard, using a pharmacological example, vascular oxidative stress can be eliminated (when restoring the functionality of eNOS) using inhibitors of the renin-angiotensin-converting enzyme, angiotensin receptor blockers, and statins. A second-generation semi-synthetic tetracycline, minocycline, can help to prevent the development of the inflammatory process in microglia (by inhibiting iNOS). Despite the limited number of studies, they have already shown perspective results in combination with basic therapy that can be applied in clinical practice, such as in treating schizophrenia and cerebral ischemia. The balance between the beneficial and toxic properties of NO in neural cells therefore varies under the influence of environmental factors and depending on the level of expression of the genes responsible for NO synthesis. These are therefore the candidate genes that increase the risk of developing neuropsychiatric diseases.

## Figures and Tables

**Figure 1 ijms-21-01604-f001:**
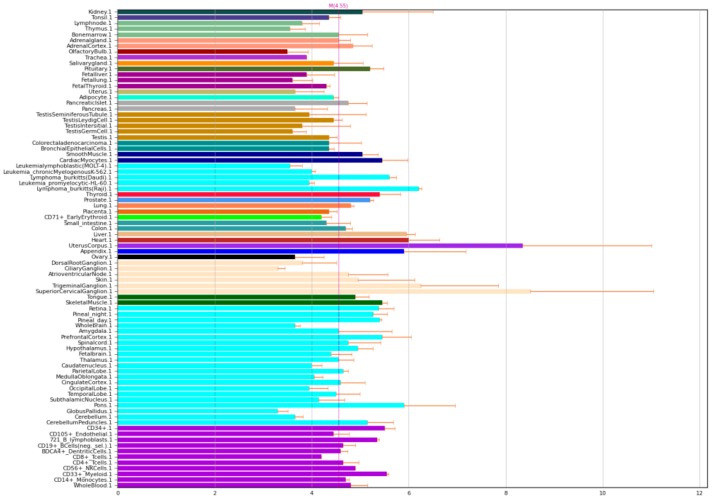
Expression of NOS1 in the human body [4,5].

**Figure 2 ijms-21-01604-f002:**
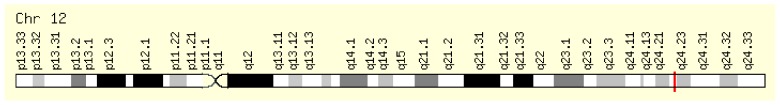
Localization of the *NOS1* gene [6,7].

**Figure 3 ijms-21-01604-f003:**
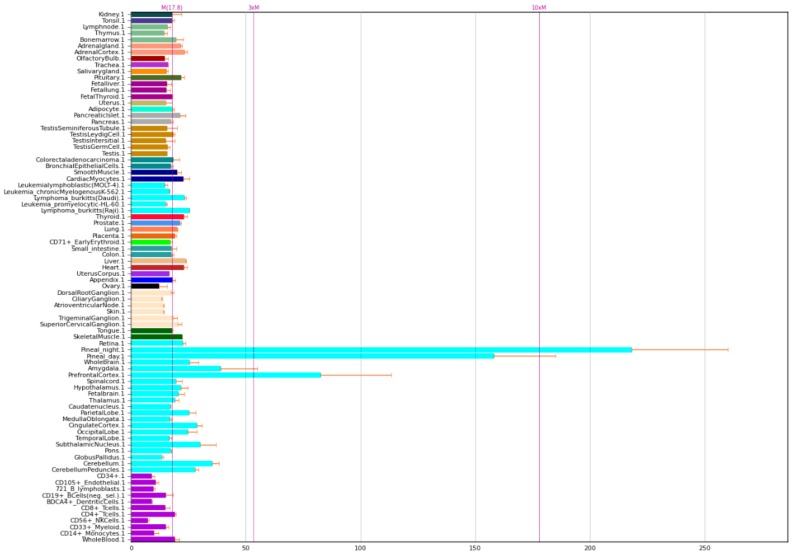
Expression of NOS1 adaptor protein in the human body [5].

**Figure 4 ijms-21-01604-f004:**
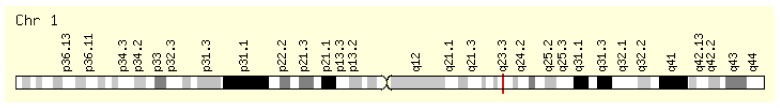
Localization of the *NOS1AP* gene [7].

**Figure 5 ijms-21-01604-f005:**
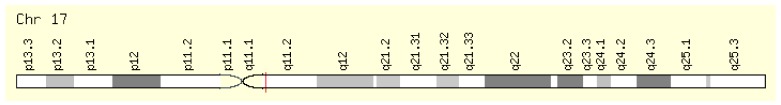
Localization of the *NOS2* gene [7].

**Figure 6 ijms-21-01604-f006:**
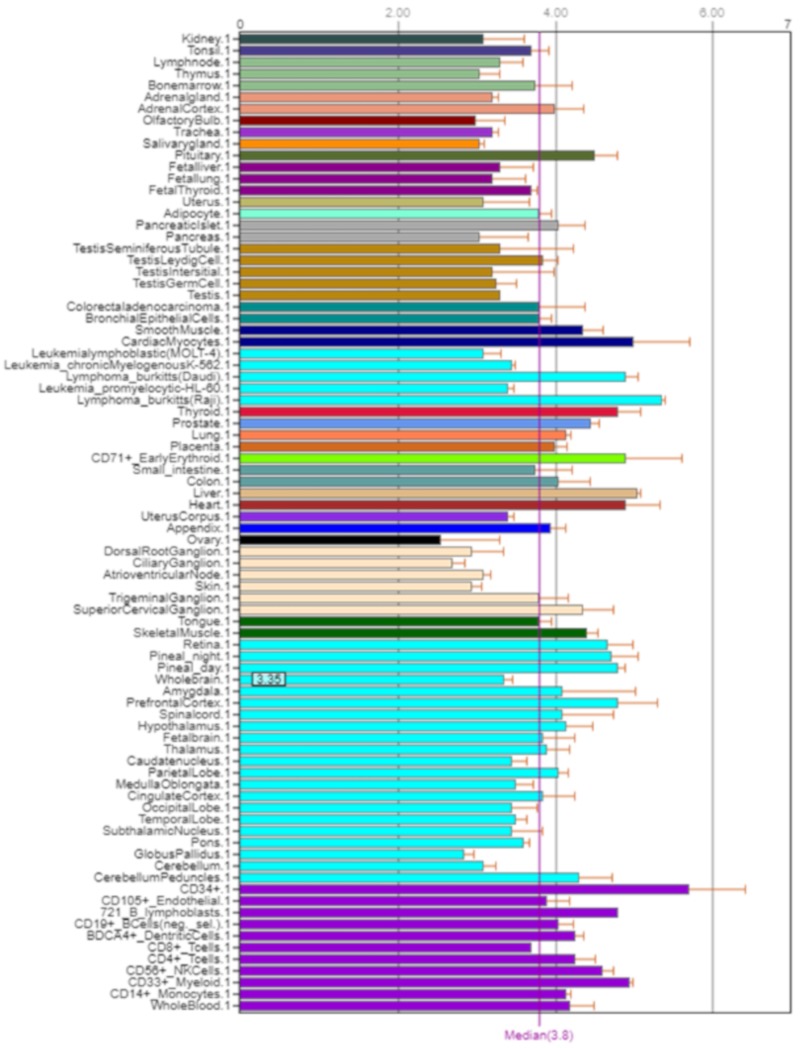
Expression of NOS2 in the human body [5].

**Figure 7 ijms-21-01604-f007:**
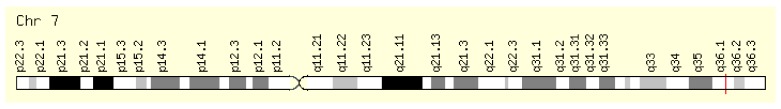
Localization of *NOS3* [7].

**Figure 8 ijms-21-01604-f008:**
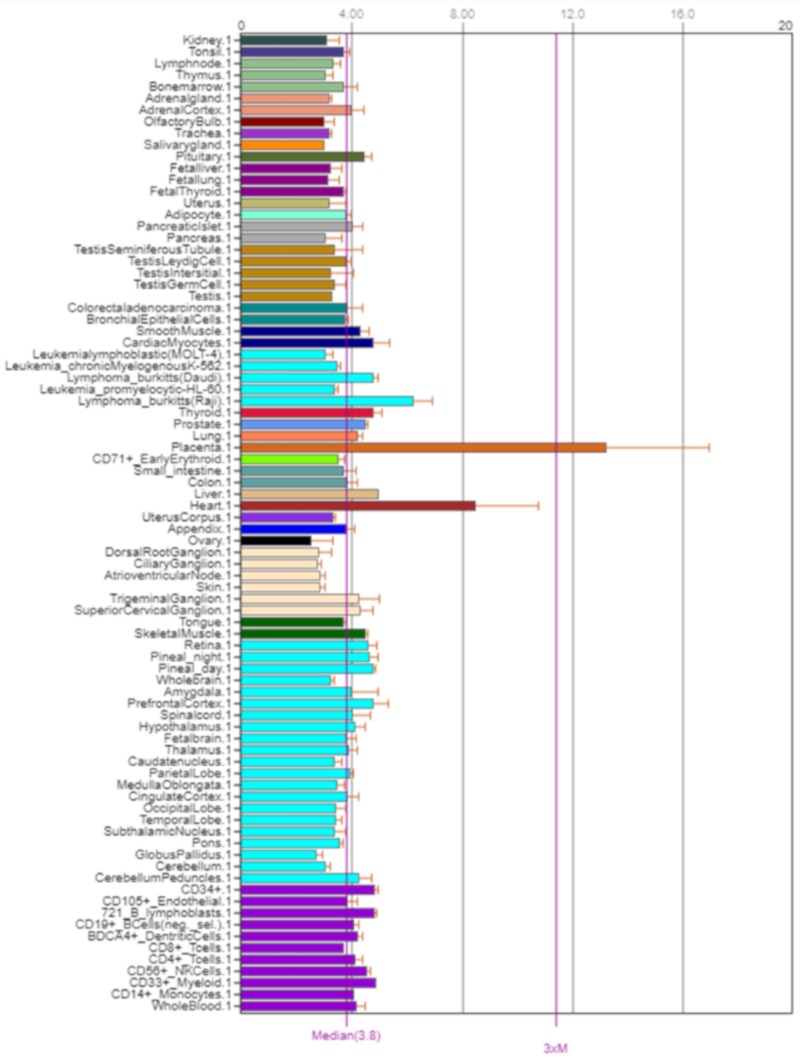
Expression of NOS3 in the human body [5].

**Table 1 ijms-21-01604-t001:** Association of NOS1 SNVs with neuropsychiatric disorders. ASD: Autism Spectrum Disorders, BAD: Bipolar Affective Disorder, MS: Multiple Sclerosis, PD: Parkinson’s Disease, RLS: Restless Legs Syndrome, SNV: Single Nucleotide Variant.

SNV	Psychiatric Disorders				Neurologic Disorders					
	Schizophrenia	Depression	BAD	ASD	PD	Brain Tumors	Stroke	Cognitive Disorders	MS	RLS
rs1004356		+ [20]								
rs1047735	– [9,11]		– [9]	– [27]	+ [3]					
rs10507279		+ [20]								– [40]
rs10774907										– [40]
rs10774910					– [3]					
rs10850803						– [30]				
rs10850807										– [40]
rs1093329										– [40]
rs1093330							– [38]			
rs11068428							– [38]			
rs11068438							– [38]			
rs11068445							– [38]			
rs11068446										– [40]
rs11068447					+ [3]					
rs1123425					– [3]	– [30]	– [38]			– [40]
rs11611788					– [3]		– [38]	+ [36]		
rs12578547					– [3]					– [40]
rs12829185					+ [3]		– [38]			
rs1353939						– [30]				
rs1483757					– [3]		+ [38] – [39]			
rs1520810		– [20]					– [38]			
rs1520811	+ [15]				– [3]					
rs1552227					– [3]		– [38]			– [40]
rs1552228					– [3]					
rs1607817							– [38]			
rs17509231							– [38]			
rs1875140		– [20]								
rs1879417	+ [16] – [10]	– [21] – [22]					– [38]	– [21]		– [40]
rs2133681	– [11]									
rs2139733							+ [38] – [39]			
rs2291908						– [30]				
rs2243044				– [27]						
rs2293048										– [40]
rs2293050		+ [20]					+ [38] – [39]			
rs2293051	– [9]		– [9]							
rs2293052					+ [3]					
rs2293054	– [11]			– [27]	– [3]					– [40]
rs2293055					– [3]					– [40]
rs2650163										– [40]
rs2682826	+ [8] - [9,10,13,14]	+ [21]	- [9,26]	- [27]	+ [3]			- [21]	- [41]	- [40]
rs3741473	- [10]									
rs3741475				- [27]	+ [3]		- [38]			
rs3741476				-[27]						
rs3741480				- [27]						
rs3782206	+ [10] – [9,13,14,16]		– [9]					+ [35]		
rs3782218		+ [20]			+ [3]		– [38]			
rs3782219	+ [9] – [10,13,14,16]		– [9]							
rs3782221	+ [9] – [10,13,14,16]		– [9]		– [3]		– [38]			
rs3837437	+* [10]									
rs41279104	+[11,14,16] – [10,13]	– [20,25]	– [25]					+ [34]	[41]	
rs473640										– [40]
rs4766836										– [40]
rs4767523					– [3]					
rs4767533		− [20]								
rs4767535						− [30]				− [40]
rs4767540	− [10,16]									
rs478597					− [3]					
rs483589						− [30]				
rs499776	+* [10] + [16]									
rs499813										– [40]
rs522910		+ [20]								
rs527590		– [20]					– [38]			– [40]
rs530393										– [40]
rs532967	– [9]		– [9]							
rs545654					– [3]	– [30]				– [40]
rs547954					– [3]		– [38]			
rs561712	+* [10] – [9,13,14]	+ [20]	– [9]		– [3]					
rs576881							– [38]			
rs579604		– [20]								
rs6490121	+ [12] – [13,14,17]					– [30]		+ [31,32,33]		– [40]
rs693534		+ [20]			– [3]					+ [40]
rs7133438										– [40]
rs7139256					– [3]					
rs7298903						– [30]	– [38]			
rs7295972					+ [3]					
rs7308402							+ [38] – [39]			
rs7309163							– [38]			
rs7314935							– [38]			
rs7959232		– [20]								
rs7977109							– [38]			+ [40]
rs816292										– [40]
rs816293						– [30]	– [38]			
rs816296		– [20]								
rs816346							– [38]			
rs816347										– [40]
rs816351						– [30]				
rs816353							– [38]			
rs816354					– [3]		– [38]			
rs816357		- [20]					- [38]			
rs816361							- [38]			
rs877995										- [40]
rs904658							- [38]			
rs9658247				– [27]						
rs9658255				– [27]						
rs9658266							– [38]			
rs9658267							– [38]			
rs9658281		+ [20]					– [38]			
rs9658536							– [38]			
rs9658570										– [40]

*—gaplotype.

**Table 2 ijms-21-01604-t002:** Association of NOS2 SNVs with neuropsychiatric disorders. IS: Ischemic Stroke.

SNV	Psychiatric Disorders		Neurologic Disorders				
	Depression	ASD	PD	Migraines	IS	Brain Tumors	MS
rs944725			– [3]			+ [30]	
rs10459953	–* [22]	– [27]					
rs1060826		+ [27]	– [3]				
rs1137933		– [27]	– [3]		– [57]		
rs2072324			+ [3]				
rs2248814			– [3]				
rs2255929		– [27]	+ [3]				
rs2297515	+* [22]		– [3]				
rs2297516			+ [3]			+ [30]	
rs2297518	– [22], +* [22]	– [27]	– [3]	+ [54,55]	– [56]		
rs2314810			– [3]				
rs2779248		– [27]					
rs2779249				+ [54], – [55]			
rs2779252						+ [30]	
rs3201742		– [27]					
rs3730014			– [3]				
rs3794764			+ [3]				
rs3794766			– [3]				
rs4795067			– [3]			+ [30]	
rs7406657		– [27]					
rs8068149		+ [27]					
rs8072199			– [3]			+ [30]	
rs10459953	– [22], +* [22]	– [27]					
rs12944039			+ [3]				
rs16966563			– [3]				
rs17722851			– [3]				
(CCTTT)n/(TAAA)n							+ [41]

*—gaplotype.

**Table 3 ijms-21-01604-t003:** Association of NOS3 SNVs with neuropsychiatric disorders.aSAH: Aneurysmal Subarachnoid Hemorrhage, GM-i.v.d.: GM-induced Vestibular Dysfunction, HIE: Hypoxic-Ischemic Encephalopathy, ICP: Infantile Cerebral Palsy.

SNV	Psychiatric Disorders			Neurologic Disorders									
	METH-i.ps	Suicide	Depression	PD	Migraines	Brain tumors	Dementia	IS	ICP	MS	HIE	GM-i.v.d.	aSAH
rs7830	– [58]												
rs743506					– [65]								
rs743507	– [58]												
rs1549758				– [3]									
rs1799983	– [58]		– [59]	– [3]	– [63,64,65]	+ [30]	+ [62]	– [56,57]	– [66]		+* [69]	+ [67]	
rs1808593											+ [69]		
rs1800779	– [58]				– [63,64]			– [57]	– [66]	+ [41]	– [68]		
rs1800783				– [3]				– [38]		– [41]	– [68,69] +* [69]		
rs1808593				– [3]							+ [69]		
rs2070744	– [58]	– [60]	– [59]		– [65]			– [56]		– [41]	+ [68]		– [70]
rs2373929								– [38]					
rs3918181								+ [61]					
rs3918186											– [69]		
rs3918188	– [58]										– [69]		
rs3918226					+ [63] – [65]			– [57]	– [66]				
rs3918227				– [3]							– [69]		
rs4496877						+ [30]							
rs10952298												– [67]	
rs12703107						+ [30]							
27bpVNTR		– [60]	– [59]		– [65]					+ [41]		– [67]	

*—gaplotype.

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
