# Peer review of "Genetic Factors of Nitric Oxide’s System in Psychoneurologic Disorders"

_ijms, 2020, doi:10.3390/ijms21051604_

Round 1
Reviewer 1 Report
Nitric oxide is a very interesting and intiguing neurotrasmitter. The choice of the article subject is needed.
The paper collects results from human researches over the last 15 years. The Authors in details tried to
discuss the involvement of NO genetic factors in neuropsychiatric disorders. Tables make it easy to review information.
I have only some suggestions:
1.NO is known to play role in the development or expression of tolerance and dependece of some substances
e.g. morphine, benzodiazepines. It would be useful to discuss this problem in the review.
2.I suggest to improve the conclusion. The presented data are contradictory but the Authors should try
to give some suggestions. At the end of the paragraphs is the lack of the summary.
3. line 330: depression is not a synonym of PTSD. It wil be clear to write in the title of the paragraph
- PTSD instead of depression.
4. Please use abbreviations when you previously explain them e.g. line 39 - NO, line 286 - nNOS, line 292 - NOS
and some others.
Author Response
Point 1: NO is known to play a role in the development or expression of tolerance and dependence of some substances e.g. morphine, benzodiazepines. It would be useful to discuss this problem in the review.
Response 1:
Thank you for your comment. We agree that this is an interesting question and we have reviewed the database. However, the results found did not quite fit into the concept of the current paper. Thus, we would like to leave it out of the present review.
Point 2: I suggest to improve the conclusion. The data presented is contradictory but the Authors should try to give some suggestions. The (review?) also lacks a summary.
Response 2:
We have considered your suggestion and re-written the conclusion. Lines 577-585:
Neuropsychiatric disorders are multifactorial in nature; that is, both endogenous and exogenous factors are the basis of their development. From this point of view, the NO system presents a significant interest. Considering the existing theories of the development of neuropsychiatric disorders (e.g., hypoxia, impaired glutamatergic neurotransmission, excitotoxicity, oxidative and nitrosative stress, thrombosis and atherosclerosis, vasospasm), NO, being directly involved in all of the chemical processes, is also involved in their biological manifestation.
NOS are expressed by many cell types of the human body, which explains the diversity of their functions.
- nNOS are expressed mainly in central and peripheral neurons, as well as some other cell types, providing a synaptic plasticity in the central nervous system, central regulation of blood pressure, relaxation of smooth muscles and vasodilation through peripheral nitrergic nerves.
- iNOS are expressed in many cells in response to lipopolysaccharides, cytokines, or other agents. iNOS generates a large amount of NO, which has a cytostatic effect on target cells of microorganisms, thereby underlining the pathophysiology of inflammatory diseases and septic shock, and are also involved in autoimmune reactions.
- eNOS are predominantly expressed in endothelial cells. They ensure the expansion of blood vessels, control blood pressure and responsible for other vasoprotective and anti-atherosclerotic effects. Many cardiovascular risk factors lead to oxidative stress and endothelial dysfunction in the vasculature.
An analysis of currently available literature demonstrated existing scientific and clinical interest in the prognostic role of polymorphisms of gene encoding enzymes involved in the synthesis of NO. However, to date, the contribution of genetic and environmental factors has not been sufficiently studied. Understanding these mechanisms can help find new approaches to pathogenetic and disease-modifying treatments.
The literature review indicates that genetic factors have a significant impact on the development of neuropsychiatric disorders. The existing contradictions can be explained by different study designs, small sample size, and various socio-geographical and ethnic factors. In order to achieve results with a high level of evidence, it is necessary to carry out inter-centre projects to provide larger samples and take into account the ethnicity of the participants. Environmental considerations should also be an integral part of every study. Epigenetic methods that reflect the influence of various factors on gene expression should be included in such projects to obtain multilateral conclusions regarding oxidative stress in the pathogenesis of neuropsychiatric disorders.
To date, a lack of comprehensive understanding of genetic predisposition and multifactorial nature of diseases does not allow us to recommend an etiological treatment. Nevertheless, from the point of view of pathogenetic disease-modifying therapy, the study of the NO system is a promising area. In this regard, using a pharmacological example, vascular oxidative stress can be eliminated (when restoring the functionality of eNOS) using inhibitors of renin-angiotensin-converting enzyme, angiotensin receptor blockers and statins. A second-generation semi-synthetic tetracycline, minocycline, can help to prevent the development of the inflammatory process in microglia (inhibit iNOS). Despite the limited number of studies, they have already shown perspective results in combination with basic therapy that can be applied in clinical practice, such as in treating schizophrenia and cerebral ischemia. The balance between the beneficial and toxic properties of NO in neural cells therefore varies under the influence of environmental factors and depending on the level of expression of the genes responsible for NO synthesis. These are therefore the candidate genes that increase the risk of developing neuropsychiatric diseases.
Point 3: line 330: depression is not a synonym of PTSD. It will be clearer to write in the title of the paragraph - PTSD instead of depression.
Response 3:
Yes, thank you, it will be better to substitute
- line 330 “Depression” to substitute for “Post-traumatic stress disorder”.
Point 4: Please use abbreviations when you previously explain them e.g. line 39 - NO, line 286 - nNOS, line 292 - NOS and some others.
Response 4:
Thank you for your comment.
We agree that it is best to
- lines 20, 39 “Nitric oxide” substituted for NO
- lines 61, 286 (285). “Neuronal nitric oxide synthase” for nNOS
- lines 65 (64), 292 (291), 332 (331), 359 (357), 455 (453). “Nitric oxide synthase” for NOS
Reviewer 2 Report
Dear Editor
I reviewed the manuscript entitled "Genetic factors of nitric oxide’s system in psychoneurologic disorders". In this review, authors have compiled the study results about the associations between different neurological disorders and variations/polymorphism in the NOS genes. This is a well designed manuscript and almost covered literature thoroughly, however, these bodies of evidence are presented in a pure narrative manner. Authors did not discuss these results in each section. Furthermore, the final conclusion section is also weakly written. in my view, discussing results, particularly, contradictory ones and providing a final conclusion could enrich manuscript.
Author Response
Point 1: I reviewed the manuscript entitled "Genetic factors of nitric oxides system in psychoneurologic disorders". In this review, the authors have compiled study results concerning the associations between different neurological disorders and variations/polymorphism in the NOS genes. This is a well-designed manuscript, and covered the literature fairly thoroughly; however, the body of evidence is presented in a purely narrative manner. The authors did not discuss these results in each section. Furthermore, the final conclusion section is also weakly written. In my view, discussing results, particularly contradictory ones, and providing a final conclusion would improve the manuscript.
Response 1:
We have considered your suggestion and re-written the conclusion. Lines 577-585:
Neuropsychiatric disorders are multifactorial in nature; that is, both endogenous and exogenous factors are the basis of their development. From this point of view, the NO system presents a significant interest. Considering the existing theories of the development of neuropsychiatric disorders (e.g., hypoxia, impaired glutamatergic neurotransmission, excitotoxicity, oxidative and nitrosative stress, thrombosis and atherosclerosis, vasospasm), NO, being directly involved in all of the chemical processes, is also involved in their biological manifestation.
NOS are expressed by many cell types of the human body, which explains the diversity of their functions.
- nNOS are expressed mainly in central and peripheral neurons, as well as some other cell types, providing a synaptic plasticity in the central nervous system, central regulation of blood pressure, relaxation of smooth muscles and vasodilation through peripheral nitrergic nerves.
- iNOS are expressed in many cells in response to lipopolysaccharides, cytokines, or other agents. iNOS generates a large amount of NO, which has a cytostatic effect on target cells of microorganisms, thereby underlining the pathophysiology of inflammatory diseases and septic shock, and are also involved in autoimmune reactions.
- eNOS are predominantly expressed in endothelial cells. They ensure the expansion of blood vessels, control blood pressure and responsible for other vasoprotective and anti-atherosclerotic effects. Many cardiovascular risk factors lead to oxidative stress and endothelial dysfunction in the vasculature.
An analysis of currently available literature demonstrated existing scientific and clinical interest in the prognostic role of polymorphisms of gene encoding enzymes involved in the synthesis of NO. However, to date, the contribution of genetic and environmental factors has not been sufficiently studied. Understanding these mechanisms can help find new approaches to pathogenetic and disease-modifying treatments.
The literature review indicates that genetic factors have a significant impact on the development of neuropsychiatric disorders. The existing contradictions can be explained by different study designs, small sample size, and various socio-geographical and ethnic factors. In order to achieve results with a high level of evidence, it is necessary to carry out inter-centre projects to provide larger samples and take into account the ethnicity of the participants. Environmental considerations should also be an integral part of every study. Epigenetic methods that reflect the influence of various factors on gene expression should be included in such projects to obtain multilateral conclusions regarding oxidative stress in the pathogenesis of neuropsychiatric disorders.
To date, a lack of comprehensive understanding of genetic predisposition and multifactorial nature of diseases does not allow us to recommend an etiological treatment. Nevertheless, from the point of view of pathogenetic disease-modifying therapy, the study of the NO system is a promising area. In this regard, using a pharmacological example, vascular oxidative stress can be eliminated (when restoring the functionality of eNOS) using inhibitors of renin-angiotensin-converting enzyme, angiotensin receptor blockers and statins. A second-generation semi-synthetic tetracycline, minocycline, can help to prevent the development of the inflammatory process in microglia (inhibit iNOS). Despite the limited number of studies, they have already shown perspective results in combination with basic therapy that can be applied in clinical practice, such as in treating schizophrenia and cerebral ischemia. The balance between the beneficial and toxic properties of NO in neural cells therefore varies under the influence of environmental factors and depending on the level of expression of the genes responsible for NO synthesis. These are therefore the candidate genes that increase the risk of developing neuropsychiatric diseases.